# Identification of developmental disorders including autism spectrum disorder using salivary miRNAs in children from Bosnia and Herzegovina

Emir Sehovic[1⊙], Lemana Spahic[1⊙], Lejla Smajlovic-Skenderagic[1⊙], Nirvana Pistoljevic[2⊙], Eldin Dzanko[2⊙], Aida Hajdarpasic[3⊙]*

1 Genetics and Bioengineering, International Burch University, Sarajevo, Bosnia and Herzegovina, 2 Education for All (EDUS), Sarajevo, Bosnia and Herzegovina, 3 Department of Medical Biology, Sarajevo Medical School, Sarajevo School of Science and Technology, Sarajevo, Bosnia and Herzegovina

⊙ These authors contributed equally to this work.
* aida.saracevic@ssst.edu.ba

## Abstract

Autism spectrum disorder (ASD) is a neurodevelopmental disorder characterized by major social, communication and behavioural challenges. The cause of ASD is still unclear and it is assumed that environmental, genetic and epigenetic factors influence the risk of ASD occurrence. MicroRNAs (miRNAs) are short 21–25 nucleotide long RNA molecules which post-transcriptionally regulate gene expression. MiRNAs play an important role in central nervous system development; therefore, dysregulation of miRNAs is connected to changes in behaviour and cognition observed in many disorders including ASD. Based on previously published work, on diagnosing ASD using miRNAs, we hypothesized that miRNAs can be used as biomarkers in children with suspected developmental disorders (DD) including ASD within Bosnian-Herzegovinian (B&H) population. 14 selected miRNAs were tested on saliva of children with suspected developmental disorders including ASD. The method of choice was qRT-PCR as a relatively cheap method available in most diagnostic laboratories in low to mid-income countries (LMIC). Out of 14 analysed miRNAs, 6 were differentially expressed between typically developing children and children with some type of developmental disorder including autism spectrum disorder. Using the most optimal logistic regression, we were able to distinguish between ASD and typically developing (TD) children. We have found 5 miRNAs as potential biomarkers. From those, 3 were differentially expressed within the ASD cohort. All 5 miRNAs had shown good chi-square statistics within the logistic regression performed on all 14 analysed miRNAs. The accuracy of 5-miRNAs model training set was 90.2%, while the validation set had a 90% accuracy. This study has shown that miRNAs may be considered as biomarkers for ASD detection and may be used to identify children with ASD along with standard developmental screening tests. By combining these methods we may be able to reach a reliable and accessible diagnostic model for children with ASD in LMIC such as B&H.

**Data Availability Statement:** All relevant data are within the paper and its Supporting Information files.

**Funding:** The research and development of this publication was supported by the United States Agency for International Development (USAID) through the Marginalized Populations Support Activity in Bosnia and Herzegovina (USAID/PPMG) to NP. The views expressed in this publication do not necessarily reflect those of USAID or the United States Government, and are the exclusive responsibility of the author and the research team. The funders had no role in study design, data collection and analysis, decision to publish, or preparation of the manuscript.

**Competing interests:** The authors have declared that no competing interests exist.

## Introduction

Autism spectrum disorder (ASD) is a neurodevelopmental disorder characterized by major social, communication and behavioural challenges. More specifically, individuals with ASD emit repetitive and restricted behavioural patterns as well as atypical social tendencies, which greatly impacts their daily functioning abilities [1].

The cause of ASD is still unclear and it is assumed that both the environmental and genetic factors together influence the risk of ASD occurrence. ASD as a part of a genetic syndrome is identified in about 10% of all ASD cases and such cases are usually associated with malformations and/or dysmorphic characteristics [2–4]. The genetic contribution to ASD is undoubtedly significant as the concordance rates for monozygotic twins is 58% for male and 60% for female, while for di-zygotic twins it is 21% for male pairs and 27% for female pairs [5]. The concordance rates also show that other factors influence the occurrence of ASD such as epigenetic regulation and environment. Epigenetic gene regulation is an essential mechanism for normal brain development [6] and anomalies in the molecules responsible for this mechanism are known to cause various neurodevelopmental disorders including autism [7–12]. Moreover, prenatal exposure to some of the substances which have been shown to increase the risk of ASD are, among others, thalidomide, misoprostol, valproic acid and chlorpyrifos [13]. Today we know that the effects of prenatal exposure to alcohol, pollution, infections and inflammations, as well as assisted deliveries, have a negative impact on later health and development of a child [14, 15].

The prevalence of ASD, all over the world, has had an increasing trend [16, 17]. The latest report on prevalence in the USA shows 1 in 59 children with ASD [18]. The worldwide prevalence of ASD is estimated to be 1–2% of the population, however, prevalence varies from country to country, mainly depending on how advanced the diagnostic system is and the availability of screening and diagnostic tools [18–22]. In order to detect and diagnose ASD, the American Academy of Paediatrics and the US Preventative Service Task Force suggest the usage of valid and reliable screening and diagnostic tools [23, 24]. Unfortunately, most of these tools are not translated, validated or financially accessible for most low and middle income countries (LMIC) where it has been estimated that 250 million children younger than 5 years are at risk of not achieving full developmental potential [1,25]. Early intervention shows significant outcomes for children with ASD which mostly depends on the early detection and diagnosis of the disorder [26]. Trends of late diagnosis have to be replaced by early diagnosis due to the short window of human brain development, because of which the best results of early intervention are achieved by the age of 5 [27,28].

Bosnia and Herzegovina (B&H) is classified as a mid-income country where parents of children with developmental disorders often encounter difficulties in getting an early diagnosis for their child. Diagnosis in B&H is usually made by a process of exclusion (usually by completing various unnecessary medical tests such as MRI, EEG etc.) and based, in most cases, on clinical evaluation. Experts from various countries point out the need for all countries to develop population-based detection, screening and evidence-based intervention for children in order to increase a chance for social inclusion of all [29]. Recently, a cheap and efficient model for diagnosing ASD emerged in the field of genetics where microRNA analysis showed promising results in distinguishing children with ASD from their typically and atypically developing peers [30]. This combined with validated screening tools could offer an affordable solution for LMIC where access to expensive diagnostic tools hinders the process of diagnosis and therefore intervention itself. Genetic analyses can be performed on samples of various origins, but for the sake of diagnosis, these sources are mostly blood, saliva or buccal swabs. Chosen source for this particular application would be saliva due to the specificity of the target

population. ASD individuals are restless and taking their blood sample or buccal swab is highly demanding even for experienced individuals. On the other hand, saliva can be collected by simple means of saliva collection which is easy to accomplish even with extreme cases of ASD and DD [31–34].

MicroRNAs (miRNAs) are a group of relatively small (around 21 nucleotides) noncoding transcripts that can modify cellular messenger RNAs (mRNAs) and protein levels by interacting with specific mRNAs. The interaction of miRNAs with mRNAs usually occurs at the 3' untranslated region (UTR) which results in mRNA degradation or repression of translation [35, 36] through partial sequence complementation [37]. It is thought that around 10–30% of all human genes could be miRNA targets [38, 39].

MiRNAs have an important role in central nervous system development and function [40, 41]. Therefore, it is not surprising that the dysregulation of miRNAs is connected to changes in behaviour and cognition observed in many neuropsychiatric disorders [42]. More specifically, miRNAs have important functions in neurogenesis, synaptogenesis and neuronal migration [43]. Usually the main function of miRNAs in these processes affect the spatial localization or compartmentalization of protein translation in different neuronal subregions, such as axons, dendrites, and synapses [44,45]. MiRNAs have been found to be dysregulated in children with ASD in various analysed biomaterials such as post-mortem cerebellar cortex [46], several post-mortem's Brodmann's areas [47,48], serum [49], peripheral blood [50,51], whole blood [52], saliva [53], olfactory mucosal stem cells [54], lymphoblast cell lines [55–57].

Successful models for diagnosing ASD utilizing various RNA molecules (miRNA, piRNA, snoRNA etc) obtained from the saliva have been created. The study whose model focused entirely on miRNAs obtained an accuracy higher than 95% [53], while the study which generated a model on various types of RNA molecules obtained a positive predictive value of 91% [58].

In order to explore affordable and quick screening models and detection of developmental disorders including ASD in B&H, we have performed a pilot study on molecular biomarkers as potential detection of atypical development. The aim of this study was to evaluate expression levels of 14 selected miRNAs from saliva of children with suspected developmental disorder and to test their ability to detect developmental disorders (DD) including ASD. The proposed panel was previously tested by Hicks et al., 2016 [53] using Next Generation Sequencing, while our method of choice was qRT-PCR, as a relatively cheap method found in most diagnostics labs in low to mid-income countries such as B&H.

## Materials and methods

### Participants and assessment

This study was approved by the EDUS Institutional Review Board for the Protection of Human Subjects (IRB) for all project activities. All participants were recruited through the Non-governmental organization EDUS-Education for All from Sarajevo, the capital of B&H. Informed written parental consent was obtained for a total of 81participants out of 126 that were enrolled in the EDUS preschool program during the school year 2018/2019. The EDUS preschool program is based on the CABAS® system (Comprehensive Application of Behavior Analysis), an evidence-based approach to assessment and treatment of Developmental Disorders and Autism Spectrum Disorder [59–63]. Children from all over B&H are enrolled in the program receiving services in developmental screening, assessment and/or treatment. One participant was excluded from the study because of the inability to collect saliva resulting in the final sample size of 80 participants consisted of children with detected developmental delays/disorders (DD) (n = 55) and typically developing (TD) pre-school children with no

previously detected developmental problems were classified as the control group (n = 25) (Table 1). The control sample was recruited from several public kindergartens for typically developing children. All data in this study were collected during the period from March to May 2018.

In order to classify participants into DD and TD groups all 80 children went through a developmental screening with the EDUS Developmental Behavioural Scales (EDUS-DBS) [64]. EDUS-DBS covers all five developmental areas: speech and communication, motor development (gross and fine), cognitive development, social-emotional development, and self-help/ adaptive skills. The EDUS-DBS screening outcome indicates, for each developmental area separately, whether the child is developing typically or atypically for their chronological age. Children who showed a developmental delay in at least one area mentioned above were selected into the group of children with DD, and children that did not show developmental delays were selected into the group of TD children. The second step was to screen the group of children detected with DD for Autism symptoms with the Childhood Autism Rating Scale, Second edition (CARS-II) [65]. The final raw score of CARS-II classifies each child into one of the three different autism symptoms severity groups: Minimal-to-No Symptoms of ASD, Mild-to-Moderate Symptoms of ASD, and Severe Symptoms of ASD. The third step consisted of clinical observations conducted by an experienced child psychiatrist in the field of ASD. By taking into account the direct observation of the child and information collected from parents about their child's developmental history, the child psychiatrist concluded two possible outcomes: ASD or DD different from ASD. The child psychiatrist had no access to the CARS-II scores of the DD sample when making conclusions.

After considering the outcomes of the CARS-II and clinical observations of the child psychiatrist, the group of children with DD was divided into two groups: children identified with ASD (n = 39) and children with other DD different from ASD (n = 16). Children identified with ASD showed CARS-II scores that fall into the Mild-to-Moderate and Severe ASD Symptoms category, and children identified with other DD (different from ASD) showed scores that fall into the Minimal-to-No Symptoms of ASD CARS-II category (Table 2). Clinical observations were conducted for 26 children (47.3%) and showed 100% concordance between Child Psychiatrists observations and CARS-II results with the CARS-II classifications (i.e. CARS-II scores indicated Mild-To-Moderate or Severe Symptoms of ASD and clinical opinion concluded ASD, CARS-II scores indicated Minimal-to-No Symptoms of ASD and Clinical opinion concluded that it is another DD different from ASD) (Table 2). Children for clinical observations from a Child Psychiatrist were selected randomly out of the whole DD sample (n = 55). No predetermined criteria for selection of participants was used.

## Molecular analysis

A total of 80 saliva samples were collected in a non-fasting state after rinsing with tap water with at least 30 minutes timespan from the last meal. Approximately 2 mLs of saliva were obtained, per manufacturer's instructions, using Samplifybio saliva collection kit (Samplfybio, Beverly, MA) and stored at room temperature until processing.

**Table 1. Characteristics of the control sample.**

| Characteristics | *n* (%) | Mean Age expressed in months |
|---|---|---|
| Male | 11 (44.0%) | 74.9 (SD = 14.4, Min = 54, Max = 95) |
| Female | 14 (56.0%) | 64.8 (SD = 12.5, Min = 42, Max = 79) |
| Total | 25 (100.0%) | 69.3 (SD = 14, Min = 42, Max = 95) |

**Table 2. Characteristics of the DD participants.**

| Characteristics | Developmental disorder | Autism Spectrum disorder |
|---|---|---|
| **Gender** | | |
| Male | 14 (87.5%) | 25 (64.1%) |
| Female | 2 (12.5%) | 14 (35.9%) |
| Total | 16 (100.0%) | 39 (100.0%) |
| **Mean age of Participants** | | |
| Male | 60.4 months (SD = 12.6, Min = 38, Max = 80) | 63.2 months (SD = 15.1, Min = 39, Max = 92) |
| Female | 61 months (SD = 5.6, Min = 57, Max = 65) | 56.5 months (SD = 14.7, Min = 37, Max = 85) |
| Total | 60.4 months (SD = 11.8, Min = 38, Max = 80) | 60.8 months (SD = 15.1, Min = 37, Max = 92) |
| **Detected developmental delay** | | |
| 1 Developmental Area | 1 (6.2%) | 1 (2.6%) |
| 3 Developmental Areas | 2 (12.5%) | 1 (2.6%) |
| 4 Developmental Areas | 3 (18.8%) | 5 (12.8%) |
| 5 Developmental Areas | 10 (62.5%) | 32 (82.0%) |
| Total | 16 (100.0%) | 39 (100.0%) |
| **CARS-II screening outcomes** | | |
| Minimal-to-No Symptoms of ASD | 16 (100.0%) | 0 (0.0%) |
| Mild-to-Moderate Symptoms of ASD | 0 (0.0%) | 12 (30.8%) |
| Severe Symptoms of ASD | 0 (0.0%) | 27 (69.2%) |
| Total | 16 (100.0%) | 39 (100.0%) |
| **Clinical opinion from Child Psychiatrist** | | |
| Male observed | 5 | 14 |
| Female observed | 1 | 6 |
| Mean age | 66.5 (SD = 3,0 Min = 62, Max = 70) | 61.9 (SD = 14.3, Min = 37, Max = 91) |
| Concluded ASD | 0 | 20 |
| Concluded other DD | 6 | 0 |
| Sample covered | 6 (37.5%) | 20 (51.3%) |

For the purpose of analysing 14 miRNAs of interest (miR-628-5p, miR-127-3p, miR-27a-3p, miR-335-3p, miR-2467-5p, miR-30e-5p, miR-28-5p, miR-191-5p, miR-23-3p, miR-3529-5p, miR-218-5p, miR-7-5p, miR-32-5p and miR-140-3p), total RNA was isolated from saliva samples using the mirVANA isolation kit (Invitrogen™), according to the manufacturer's instructions. The purity of isolated RNA was determined by OD260/280 using a Nanodrop (Thermo Scientific, Worcester, MA).

The miRNAs of interest were reversely transcribed using TaqMan MicroRNA Reverse Transcription Kit (Applied Biosystems™ Foster City, CA) and specific miRNA reverse transcription (RT) primers, on GS1 Thermal Cycler System (G-Strom) and SimpliAmp Thermal Cycler (Applied Biosystems™ Foster City, CA) according to the manufacturer's instructions.

After reverse transcription, due to smaller amounts of RNA within the samples, pre-amplification was performed. For pre-amplification, a custom primer pool was prepared with 14 miRNAs of interest. The total volume of the primer pool contained 70 μL of TaqMan Micro-RNA Assay + 430 ddH$_2$0 for a total of 500 μL. In the mastermix for the pre-amplification reaction, for each sample, we used 1.9 μL of pre-amplification primer pool, 0.13 μL of AmpliTaq Gold, 0.75 μL of MgCl, 1.25 μL of 10x PCR Buffer Gold, 1.0 μL of dNTPs and 6.22 μL of ddH$_2$0.

The quantification of targeted miRNAs was performed using TaqMan MicroRNA Assay (Applied Biosystems™ Foster City, CA) on Agilent (Stratagene, La Jolla, CA) MX3005P Multiplex QPCR Real-time Thermal Cycler. For the purpose of quantification of targeted miRNAs,

a mastermix was made with 5 μL of TM Universal mastermix, 0.5 μL of 20x TM Micro RNA Array assay for a particular miRNA and 4.30 μL of ddH$_2$0. The total mastermix volume was 9.80 μL. The qPCR reaction was performed on all miRNAs individually. For each miRNA the appropriate 20x TM Micro RNA Array assay was used. The dye used for miRNA comparative quantitation was FAM. Within each plate a calibrator well was selected for the purpose of normalization of data. Data was organized for each miRNA and necessary replicates were performed.

## Statistical analysis

Normalization of expression data was performed using Delta Delta Ct, as suggested by [66], which is calculated according to formula: (original Ct value of a sample-calibrator value of the miRNA)—(calibrator of the miRNA from the sample–calibrator of the normalizing miRNA). The normalizing miRNA used in this study was miR-191-5p [67].

Outliers were determined using the Grubbs test via Xlstat add-on with Microsoft Office Excel program. Since obtained results have normalized Ct values, criteria for the classification of an outlier was used. Hence, all normalized expression data values with a z-score over 1.5, for both tails, were classified as outliers. The 1.5 z-score threshold was chosen because the Ct values above that threshold are too high or too low in order to be considered valid [68]. The data was tested for normal distribution using Shapiro Wilk, Anderson-Darling and Jarque-Bera test. The normal distribution tests were performed in the Past 3 program [69] (https://folk.uio.no/ohammer/past/).

Normally distributed data from qRT-PCR were analysed using Student's T-test while the not-normally distributed data was analysed using Mann-Whitney U test. Logistic regression analysis was used to describe the dependency of the dependent outcome, the diagnosis in this case, and was performed on each miRNA separately. In addition, a logistic regression was performed on multiple sets of variables (miRNAs) in order to predict the diagnosis (ASD or DD). The logistic regression was calculated with Xlstat add-on within the Microsoft Office Excel program.

A Receiver Operating Characteristic (ROC) curve of the created prediction models was also calculated as it can provide an unbiased assessment of the overall model performance. Therefore, the ROC can be used for evaluating the diagnostic power of miRNAs. In order to further support the results found using the logistic regression a multivariate linear regression of Partial Least Squares-Discriminant analysis (PLS-DA) was calculated. The PLS-DA calculates the multivariate linear combinations from the 14 miRNAs of interest that are best predictors of the class of interest. PLS-DA was also calculated with the Xlstat add-on. For all models, a cross validation was performed by randomly dividing the dataset into a training and validation set. The size of training and validation sets were determined based on the sample size being analysed (approximately 10%). Multiple iterations of the models were performed to ensure randomness within the models. Furthermore, parameters chosen were the default parameters (0.5 prediction threshold) and the sporadic missing data was predicted using the nearest neighbour method approach.

## Results

Out of 14 analysed miRNAs, 6 were differentially expressed between typically developing children and children with some type of developmental disorder (which includes ASD). These were miR-7-5p, miR-23a-3p, miR-32-5p, miR-140-3p, miR-628-5p and miR-2467-5p. Two miRNAs were up-regulated (miR-7-5p and miR-2467-5p), while 4 were down-regulated (miR-23a-3p, miR-32-5p, miR-140-3p and miR-628-5p) (Table 3). MiRNA with largest difference in

average expression was miR-32-5p, followed by miR-23a-3p. Normalized Ct value data for each individual miRNA within the three cohorts was presented in a Box Plot and Jitter graph (Fig 1). Details on the percentage of obtained Ct values for every cohort and for each individual miRNA can be seen in S1 Table.

Out of 14 miRNAs, analysed in ASD samples, 5 were differentially expressed according to Mann Whitney U test. Four miRNAs were down-regulated while one was up-regulated (Table 3). Down-regulated miRNAs within ASD group were miR-23a-3p, miR-32-5p, miR-628-5p and miR-140-3p, while significantly up-regulated was miR-7-5p. MiRNA with the largest difference in average expression was miR-32-5p, followed by miR-23a-3p. Differentially expressed miRNAs within non-ASD group, when compared to the control group, were miR-23a-3p, miR-32-5p, mir-628-5p and miR-2467-5p. MiR-2467-5p was upregulated while the other 3 miRNAs were all down-regulated (Table 3).

Logistic regression performed on individual miRNAs between TD and children with ASD, as a subgroup of DD, has shown that the best performing miRNAs in differentiating between these groups were miR-32-5p (-2Log(Likelihood): 7.363; p-value: 0.007) and miR-23a-3p (-2Log(Likelihood): 5.406; p-value: 0.020). Following these two miRNAs, the best performing were miR-7-5p, miR-218-5p, miR-27a-3p and miR-628-5p, although their GoF statistics were not significant. When the values of ROC AUC were taken into consideration the best performing miRNAs were miR-23-3p, miR-32-5p and miR-7-5p.

Logistic regression was also performed on individual miRNAs between typically developing children and children with some type of developmental disorder (this cohort included ASD and non-ASD DD groups) and the best performing individual miRNA in classifying between these groups were miR-32-5p (-2Log (Likelihood): 11.208; p-value: 0.001) followed by miR-23a-3p (-2Log(Likelihood): 9.507; p-value: 0.002). MiR-628-5p, miR-7-5p and miR-27a-3p have also shown a very good performance in classifying between TD and DD children. Out of the latter 3 miRNAs, only miR-628-5p had significant -2Log(Likelihood) statistic (-2Log

**Table 3. Differentially expressed miRNAs within the analysed cohorts and their directional expression changes relative to the control group.**

| Cohorts | Expression pattern (p-value) |
| --- | --- |
| **DD** | |
| miR-7-5p | Up-regulated (0.0361) |
| miR-23a-3p | Down-regulated (0.0001) |
| miR-32-5p | Down-regulated (0.0001) |
| miR-140-3p | Down-regulated (0.0067) |
| miR-628-5p | Down-regulated (0.0001) |
| miR-2467-5p | Up-regulated (0.0499) |
| **ASD** | |
| miR-7-5p | Up-regulated (0.0172) |
| miR-23a-3p | Down-regulated (0.0001) |
| miR-32-5p | Down-regulated (0.0001) |
| miR-140-3p | Down-regulated (0.0053) |
| miR-628-5p | Down-regulated (0.0005) |
| **Non-ASD DD** | |
| miR-23a-3p | Down-regulated (0.0033) |
| miR-32-5p | Down-regulated (0.0102) |
| miR-628-5p | Down-regulated (0.0135) |
| miR-2467-5p | Up-regulated (0.0057) |

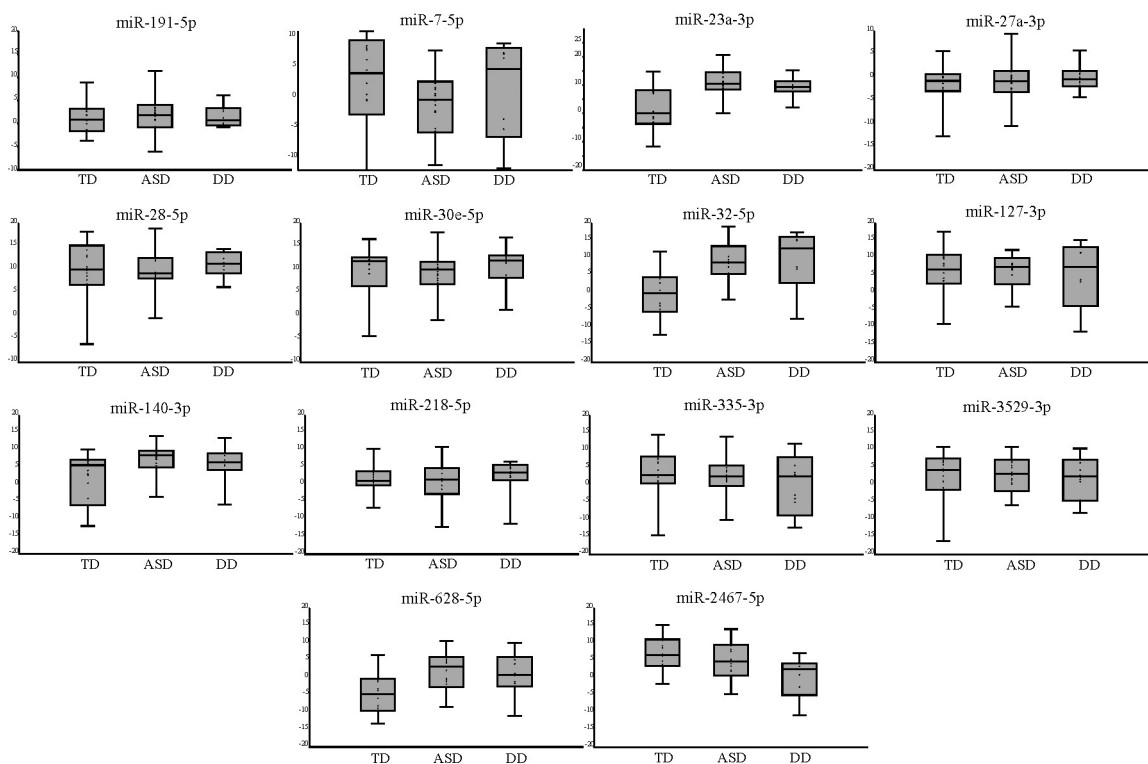

**Fig 1. Box and jitter graph of the normalized Ct values for each miRNA within the three analysed cohorts (TD, ASD and non-ASD DD).**

(Likelihood): 6.247; p-value: 0.012). When the values of ROC AUC were taken into consideration, miR-32-5p and miR-23a-3p again had the best performance. MiR-628-5p also had a relatively high ROC AUC (0.696).

In the logistic regression on individual miRNAs between non-ASD DD group and typically developing children only one, miR-32-5p, had significant -2Log(Likelihood) statistic of 9.301 with a p-value of 0.002. MiRNAs close to having a statistically significant GoF statistics were miR-23a-3p (-2Log(Likelihood): 3.127; p-value: 0.067) and miR-3529-3p (-2Log(Likelihood): 2.915; p-value: 0.088). For further details please refer to S8–S11 Tables.

PLS-DA performed on TD children and children with ASD had shown that the most contributing miRNAs in creating the PLS-DA were miR-23a-3p, miR-32-5p, miR-7-5p, miR-27a-3p, miR-628-5p and miR-140-3p, in order. The sensitivity and specificity for this PLS-DA model were 64% and 88.57% respectively. The total accuracy was 78.33%. The AUC for the ROC of the PLS-DA model was 0.920. For the PLS-DA performed on TD and DD cohorts, miRNAs with the most significant contribution to PLS-DA model were (from most to least significant): miR-23a-3p, miR-7-5p, miR-32-5p, miR-27a-3p, miR-628-5p, miR-140-3p and miR-2467-5p. The overall prediction model for the PLS-DA had a 36% specificity, 86.27% sensitivity and a total accuracy of 69.74%. For the calculated ROC Curve the AUC was 0.793. MiRNAs with the highest VIP scores were miR-23a-3p, miR-7-5p and miR-32-5p. The PLS-DA on non-ASD DD group and typically developing children had shown that the most contributing miRNAs in creating this PLS-DA prediction model were miR-32-5p, miR-23a-3p, miR-628-5p, miR-7-5p. This PLS-DA model had 88% specificity and 64.29% specificity. The total accuracy and ROC AUC were 79.49% and 0.749, respectively (S6 and S7 Tables).

**Table 4. Performance of the logistic regression prediction models (includes sensitivity, specificity and accuracy along with their confidence intervals) which have shown statistically significant ability of differentiating between the analysed groups.**

| Model | Sensitivity (95% CI) | Specificity (95% CI) | Accuracy (95% CI) |
|---|---|---|---|
| TD-DD | 86.36% (65.09%–97.09%) | 93.18% (81.34%–98.57%) | 90.91% (81.26%–96.59%) |
| TD-ASD | 90.32% (63.66%–96.95%) | 90% (77.93%–99.18%) | 90.2% (78.59%–96.74%) |
| TD—non-ASD DD | 75% (66.27%–95.81%) | 92% (57.19%–98.22%) | 85.37% (70.83%–94.43%) |

*CI—Confidence Interval.

Overfitting was suspected on the logistic regression model of 14 miRNAs which predicted ASD children from TD. That model had maximum specificity and sensitivity for the training set and 66.67% specificity and 100% sensitivity for the validation set. Hence, a more reliable model was created which included 5 miRNAs (best performing in the previously mentioned model) (miR-7-5p, miR-23a-3p, miR-27a-3p, miR-140-3p and miR-2467-5p). The -2Log(Likelihood) value of the model was 40.237 with a p-value of <0.0001. Specificity was 90.00%, sensitivity was 90.32% (Table 4) and the ROC AUC of the model was 0.952. The validation set had an accuracy of 90%.

The logistic regression on the full set of variables performed on typically developing children and children with any type of developmental disorder had a maximum specificity and sensitivity for the training set and 50% specificity and 100% sensitivity for the validation set. Since overfitting was suspected, a model with 7 miRNAs (best performing in the previously mentioned model) (miR-7-5p, miR-23a-3p, miR-27a-3p, miR-32-5p, miR-140-3p, miR-628-5p and miR-2467-5p) was created. The -2Log(Likelihood) for this model was 62.805 with a p-value of <0,0001. The specificity and sensitivity values were 86.36% and 93.18%, respectively (Table 4). The ROC AUC of the model was 0.983. The validation set had an accuracy of 90%. As the logistic regression model on non-ASD DD and TD children also had maximum accuracy, overfitting had to be taken into consideration. Hence, the best performing model on reduced number of miRNAs was based on 5 miRNAs (best performing in the previously mentioned model) (miR-7-5p, miR-23a-3p, miR-32-5p, miR-140-3p, miR-3529-3p). The -2Log (Likelihood) value of the model was 30.013 with a p-value of <0.0001. Specificity was 92.00%, sensitivity was 75.00% (Table 4) and the ROC AUC of the model was 0.940. The validation set was too small (5 samples) for meaningful conclusions. However, within randomly chosen 5 validation samples 100% accuracy in their classification was observed. Details on the mentioned models can be seen in Fig 2. Furthermore, we have performed a power analysis (where the number of final sample sizes for each miRNA or test can be seen) on all models and individual miRNAs which have shown statistically significant ability in differentiating between cohorts (S2–S5 Tables).

## Discussion

We have analysed expression of 14 miRNAs in saliva samples of children with DD including ASD and TD. Six miRNAs were found to be differentially expressed in children with DD. Furthermore, when samples of children with ASD were analysed, 5 miRNAs were found to be differentially expressed. All 5 miRNAs from ASD analysis were found within the group of 6 miRNAs differentially expressed in children with DD. Only miR-2467-5p was not differentially expressed among children with ASD. We have found that this particular miRNA had significant difference in abundance in ASD when compared to other DD samples. Out of differentially expressed miRNAs among children with ASD, with the exception of miR-140-3p and miR-628-5p, all had concordant expression patterns to a previously reported study [53].

A) DD and TD

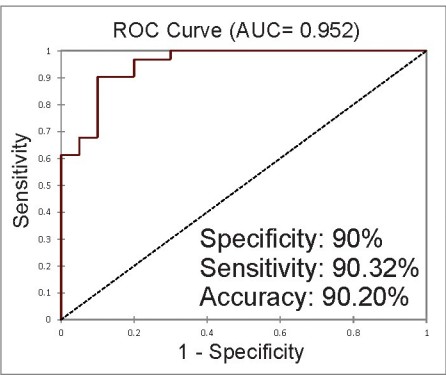

B) ASD and TD

C) Non-ASD DD and TD

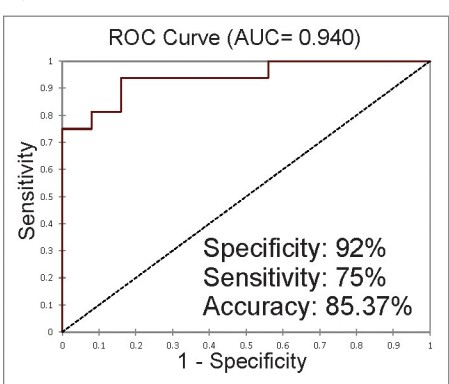

D) ASD and non-ASD DD

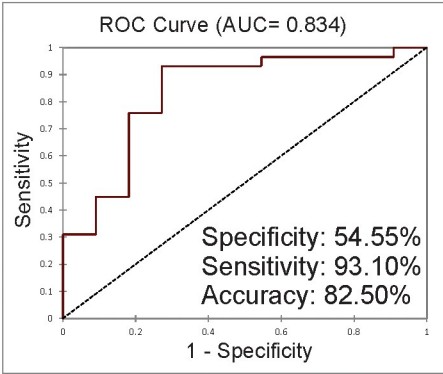

**Fig 2. Comparison of the 4 logistic regression models.** Models A, B and C have shown statistically significant differentiation between groups. Presented are ROC AUC, sensitivity, specificity and accuracy values of the models.

In our study miR-140-3p and miR-628-5p were down-regulated, while they were up-regulated in the mentioned study.

Using the most optimal logistic regression model, we were able to distinguish between ASD and TD children. We have found 5 miRNAs as potential biomarkers. Out of these 5 miRNAs, 3 were differentially expressed within the ASD cohort. All 5 miRNAs have shown good chi-square statistics within the logistic regression model which utilized all 14 miRNAs analysed in this study. The accuracy of 5-miRNA logistic regression model training set was 90.20%, while the validation set had a 90% accuracy. The logistic regression performed only on differentially expressed miRNAs between ASD and TD has shown poorer results than the previously discussed model. Moreover, the best logistic regression model which attempted to distinguish between DD and TD children included 7 miRNAs as variables (miRNAs). Six of these were differentially expressed among children with DD while one, which was not differentially expressed (miR-27-3p), has shown a relatively large VIP value within the PLS-DA. This model had a total accuracy of 90.91% in the training set and 90% accuracy within the validation set. The best performing logistic regression model, attempted to differentiate between non-ASD DD and TD cohorts, included 5 variables. Two of the five miRNAs were differentially expressed, while 3 have performed very well in the model which included the complete set of 14 miRNAs. In this study cross validation was used for validating the trained models. In order to obtain more reliable validation results an independent sample cohort for validation would be suitable.

The three main prediction models (distinguishing children with ASD, any type of DD or non-ASD developmental disorder from TD) have all shown results with significant GoF statistics and satisfactory accuracy. The model used to differentiate between children with DD and TD had the best performance followed by the model on ASD and TD children. Finally, the model which attempted to differentiate between non-ASD DD and TD children had the poorest performance out of the 3 models. The results obtained from the discussed logistic regressions models were fully supported by the PLS-DA performed on the same groups.

In this study we have found that, individually, the best miRNAs for differentiating between children with ASD or even DD from TD were miR-23-3p and miR-32-5p. The same was true when differentiating non-ASD cohort from TD. It has been found that miR-23a-3p functions cooperatively with miR-27a-3p to regulate cell proliferation and differentiation [70]. Furthermore, miR-23a-3p and miR-27a-3p were also dysregulated in a number of human diseases and disorders, including ASD [49, 53, 71]. Levels of miR-23a-3p also fluctuated in response to CNS injuries such as cerebral ischemia [72] or temporal epilepsy [73], both of which are associated with ASD [74]. MiR-23a-3p was also found to be dysregulated in 3 other studies [42, 45, 49]. MiR-32-5p is believed to be involved in certain processes which promote cell proliferation, migration and suppresses apoptosis in breast cancer while inhibiting proliferation and invasion in gastric cancer cell lines [75, 76].

Studies on biomarkers, as potential diagnostic tools for ASD, have also developed successful prediction models. There have been numerous studies on differentially expressed miRNAs within ASD children including analysis of their suitability as biomarkers [51]. MiR-23a-3p analysed by Hicks et al., 2016 [53], Sarachana et al., 2010 [55] has been shown to be significantly down-regulated. Moreover, when it comes to individual performance of miRNAs, differentially expressed miRNAs in this study had a lower prediction power than in Hicks et al., 2016 [53] or Vasu et al., in 2014 [49]. Prediction models utilizing multiple miRNAs or other types of RNAs as biomarkers for ASD have also been studied and overall show better prediction capabilities. One such example is the study which utilized 14 miRNAs and obtained 95.6% specificity, 100% sensitivity and an area under the ROC curve of 0.974 [53]. In this study, the best individually performing miRNAs were miR-335-3p and miR-30-5p, whereas in our study the best performing miRNAs were miR-23a-3p and miR-32-5p.

In addition, another study [58] used a combination of RNA molecules (1 snoRNA, 8 piRNAs, 4 precursor miRNAs, 7 mature miRNAs and 12 microbial taxa) and reported 78.3% specificity, 79.9% sensitivity and an area under the ROC curve of 0.868.

When analysing miRNAs in ASD individuals relatively small percentage of differentially expressed miRNAs overlap between studies. A total of 3 miRNAs showed consistent dysregulation in 3 or more studies [77]. These differences in obtained differentially expressed miRNAs, also applicable to our study, can be explained by the effect of different factors such as age, ethnicity, ASD heterogeneity, different RNA collection devices, RNA quantification and analysis and miRNA role in development. It has been shown that the overall ASD prevalence is higher in males then in females, [78, 79] as was observed in our study. One potential explanation [80] is the female genome resistance and the fact that more severe symptoms are usually required in order for females to be detected [78–81]. As such biases are present across biomarker ASD studies [43, 58], more research would have to be performed in order to determine the effect gender has, if at all, on miRNA expression. The referenced study [53] involved children 5 to 14 years of age. In our study, on the other hand, we involved children (Table 2) 3 to 8 years of age in an attempt to test children as early as possible or as close as possible to ASD diagnosis in order to evaluate the combination of performed tests (clinical and biomarker) and their accuracy in diagnosing ASD. Furthermore, a recent review reported no differences in prevalence across geographic regions or variability based on ethnicity or socioeconomic factors; however, they did add that the lack of

comprehensive datasets from low-income countries impacts the ability to detect these effects [16]. Therefore, testing miRNAs as biomarkers of ASD in different ethnic groups including those from LMIC's is absolutely needed. We have found correlation with compared studies as well; however, further model improvement has to be performed on much larger population. In addition, Hicks' study targeted children with "high functioning" ASD (average ADOS-II score = 10.6 ± 4.1), whereas our study included ASD and DD children classified by CARS-II as Mild-to-Moderate and Severe ASD Symptoms category for ASD and Minimal-to-No Symptoms of ASD CARS-II category for non-ASD DD children. Because salivary miRNA expression is associated with levels of ASD symptoms, it is likely that this also contributes to observed differences in miRNA expression. In order to validate used scale (CARS II) we have randomly selected 26/55 (47%) children for clinical evaluation by child psychiatrist. This validation showed 100% concordance between CARS II and child psychiatrists' evaluation and confirmed the reliability of the used scale. However, the fact that we performed clinical evaluation on approximately 50% of children may represent a limiting factor on such a validation. In our study we used collection devices (Samplifybio saliva collection kit) different from the referred study [53]. The RNA preservation liquid between the two devices may have, additionally, contributed to miRNA expression differences. Our method of choice was RT-PCR as a relatively cheap and available method in most medical labs in LMIC. Therefore, we expected differences in expression and detection obtained by RT-PCR when compared to a more robust sequencing method used by Hicks et al. 2016. Nonetheless, models utilizing salivary miRNA and detection with RT-PCR can be used to differentiate children with ASD from typically developing or non-ASD DD children [30]. However, in order to improve their specificity and make them applicable in clinical setting, the proposed models have to be tested on a larger population and potentially, as suggested by Hicks et al. 2018, by employing a multi-"omic" approach using additional RNA families.

Furthermore, as miRNA expression is a dynamic process which changes throughout development we have to consider its effect on miRNA expression differentiation [43] However, it has been shown that miRNAs are essential for survival and differentiation of newborn neurons but not for expansion of neural progenitors during early neurogenesis in the mouse embryonic neocortex [43]. Although there is a chance that some of the found differences are affected by the miRNA involvement in neural development, majority of our findings are on children of average age of 5 and same ethnicity. Looking at the younger children such as infants and toddlers may shed more light in the future on the potential differences in miRNA expression due to developmental processes better.

Finally, in an earlier *in silico* study [82] target mRNAs of the same 14 miRNAs were analysed. Genes reported to be targeted by all 14 miRNAs and have more than 7 predicted target sites were: MAPK10, KCNMA1 and DST. The number of predicted sites was used as an indirect measure of how correlated an mRNA was with the miRNA in question. In the same study, genes reported to be targeted by 13 miRNAs and have more than 7 predicted target sites were reported. Those genes were ZBTB20, GAS7, NTRK2 and SCN2A. It is thought that most of these genes are related to neural processes and are directly or indirectly related to ASD. According to Hicks et al., 2016 [49] the notable ASD-associated mRNA targets of these 14 miRNAs are Fragile X Mental Retardation (FMR1) and Forkhead Box Protein P2 (FOXP2). Those mRNAs predicted to have a function most relevant to ASD were mapped to the Neuron Projection and Axon Projection subnodes.

## Conclusion

This study has shown that miRNAs can be considered as biomarkers for ASD diagnosis and could be used to identify children with ASD at a very early stage of life. We performed a

transdisciplinary cooperation in order to define the most optimal and accurate approach in identifying children with developmental disorders, including ASD, in low to mid income countries such as Bosnia and Herzegovina. We have done this by utilizing a combination of molecular analysis, based on miRNAs as biomarkers, and screening methods such as EDUS-DBS, CARS II and clinical analysis. Our molecular analysis was based on a panel of 14 miRNAs previously shown as good biomarkers for ASD. We have found that even subsets of this panel of miRNAs have the potential to be used as diagnostic biomarkers for ASD and/or DD. Out of 14 miRNAs analysed in this study, miR-32-5p, miR-23a-3p and miR-7-5p, have been found as good candidates for biomarkers in differentiating children with ASD from typically developing children. The best miRNAs for differentiating between children with any type of developmental disorder and typically developing children were miR-23a-3p, miR-32-5p and miR-628-5p. As previously suggested, a good step towards implementing miRNAs as ASD or DD biomarkers would be large scale validation study with multi-"omics" approach along with currently employed screening tests. This would lead to an optimized biomarker diagnostic tool that complements current screening tests.

## Supporting information

**S1 Table. Percentage of obtained Ct values for every cohort and for each individual miRNA.**
(DOCX)

**S2 Table. Shown is the detailed logistic regression performance of individual miRNAs between TD and DD cohorts.**
(DOCX)

**S3 Table. Shown is the detailed logistic regression performance of individual miRNAs between TD and ASD cohorts.**
(DOCX)

**S4 Table. Shown is the detailed logistic regression performance of individual miRNAs between TD and non-ASD DD cohorts.**
(DOCX)

**S5 Table. Shown is the detailed logistic regression performance of individual miRNAs between ASD and non-ASD DD cohorts.**
(DOCX)

**S6 Table. Details on performance of the PLS-DA prediction models on the analysed cohorts.**
(DOCX)

**S7 Table. MiRNAs with Variable Importance in Projection (VIP) value over 1.0 for each cohort within the PLS-DA.**
(DOCX)

**S8 Table. Power analysis (where the number of final sample sizes for each miRNA or test can be seen) on logistic regression which have shown statistically significant ability in differentiating between cohorts.**
(DOCX)

**S9 Table. Power analysis on Mann-Whitney U test on TD–DD cohorts.**
(DOCX)

**S10 Table. Power analysis on Mann-Whitney U test on TD–ASD cohorts.**
(DOCX)

**S11 Table. Power analysis on Mann-Whitney U test on TD–non-ASD DD cohorts.**
(DOCX)

**S1 File.**
(XLSX)

## Author Contributions

**Conceptualization:** Lejla Smajlovic-Skenderagic, Nirvana Pistoljevic, Eldin Dzanko, Aida Hajdarpasic.

**Data curation:** Emir Sehovic, Lemana Spahic, Nirvana Pistoljevic, Eldin Dzanko, Aida Hajdarpasic.

**Formal analysis:** Emir Sehovic, Lemana Spahic, Lejla Smajlovic-Skenderagic, Eldin Dzanko, Aida Hajdarpasic.

**Funding acquisition:** Nirvana Pistoljevic, Aida Hajdarpasic.

**Investigation:** Emir Sehovic, Lemana Spahic, Aida Hajdarpasic.

**Methodology:** Emir Sehovic, Lemana Spahic, Lejla Smajlovic-Skenderagic, Eldin Dzanko, Aida Hajdarpasic.

**Project administration:** Nirvana Pistoljevic, Aida Hajdarpasic.

**Resources:** Aida Hajdarpasic.

**Supervision:** Lejla Smajlovic-Skenderagic, Aida Hajdarpasic.

**Validation:** Aida Hajdarpasic.

**Visualization:** Emir Sehovic.

**Writing – original draft:** Emir Sehovic, Lemana Spahic, Nirvana Pistoljevic, Eldin Dzanko, Aida Hajdarpasic.

**Writing – review & editing:** Lejla Smajlovic-Skenderagic, Nirvana Pistoljevic, Aida Hajdarpasic.

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
