## [Decision Letter · Decision Letter 0]

20 Dec 2019

PONE-D-19-31049

Identification of developmental disorders including autism spectrum disorder using salivary miRNAs in children from Bosnia and Herzegovina

PLOS ONE

Dear Dr. Hajdarpasic,

Thank you for submitting your manuscript to PLOS ONE. After careful consideration, we feel that it has merit but does not fully meet PLOS ONE’s publication criteria as it currently stands. Therefore, we invite you to submit a revised version of the manuscript that addresses the points raised during the review process.

We would appreciate receiving your revised manuscript by Feb 02 2020 11:59PM. To enhance the reproducibility of your results, we recommend that if applicable you deposit your laboratory protocols in protocols.io, where a protocol can be assigned its own identifier (DOI) such that it can be cited independently in the future. For instructions see: http://journals.plos.org/plosone/s/submission-guidelines#loc-laboratory-protocols

We look forward to receiving your revised manuscript.

Kind regards,

Lucia Billeci

Academic Editor

PLOS ONE

Journal Requirements:

2. We note you have included a table to which you do not refer in the text of your manuscript. Please ensure that you refer to Table 3 in your text; if accepted, production will need this reference to link the reader to the Table.

Reviewers' comments:

Reviewer's Responses to Questions

**Comments to the Author**

1. Is the manuscript technically sound, and do the data support the conclusions?

Reviewer #1: No

Reviewer #2: Partly

Reviewer #3: Yes

Reviewer #4: Yes

2. Has the statistical analysis been performed appropriately and rigorously? 

Reviewer #1: No

Reviewer #2: Yes

Reviewer #3: Yes

Reviewer #4: Yes

3. Have the authors made all data underlying the findings in their manuscript fully available?

Reviewer #1: No

Reviewer #2: No

Reviewer #3: Yes

Reviewer #4: Yes

4. Is the manuscript presented in an intelligible fashion and written in standard English?

Reviewer #1: Yes

Reviewer #2: Yes

Reviewer #3: Yes

Reviewer #4: Yes

5. Review Comments to the Author

Reviewer #1: 1) It is unclear how the 14 miRNAs were selected. It seems to be based on reference 49 (A comparative review of microRNA expression patterns in autism spectrum disorder. Frontiers in psychiatry. 2016); however, I do not see much overlap with the 27 miRNAs discussed in that paper.

2) “The validation set” was mentioned, but it is unclear what that is.

Reviewer #2: Overall, this is an interesting manuscript which interrogates an emerging biomarker family in a clinical disorder with a great need for an objective, biologic diagnostic aid.

The novelty of the manuscript lies in its unique ethno-geographic cohort, and the potential to confirm previously-published, non-invasive biomarkers in that cohort. The authors have done a thorough job providing clinical characteristics of the participants, and the statistical approach appears sound.

My main concern is regarding the interpretation of the results, and the way the discussion frames them (in the context of the existing literature). The authors are missing a key citation, which involves the largest study of saliva microRNA in children with autism (PMID: 30926572). This study showed that saliva microRNA alone does not provide sufficient accuracy for differentiating children with ASD from peers with DD or TD. This is why additional studies (e.g. citation 54) have moved to poly-omic analysis of both microRNA and other human and microbial RNAs. For this reason, a study of <100 children is not really a validation study - especially since the authors have used different saliva collection kits, different RNA extraction techniques, and different RNA quantification protocols. The discussion does not include any mention of these factors. There is also no discussion of the study's limitations.

Here are a few additional specific observations/suggestions:

The intro would benefit from a discussion of the advantages to saliva relative to blood collection.

This study does not show that microRNAs can be used as early as birth for ASD diagnosis. There are studies to show this type of test is not even a priority for parents of children with ASD (PMID: 30903561).

I worry that the sample size (n=25 for controls) does not support separate training/test sets. A power analysis would be useful here.

Table 1 needs to compare sex/age across groups (were there differences?). Also, details about diet, oral hygiene, and oral meds would help with interpretation of the findings.

Were any participants excluded for inability to expectorate?

Please provide data on RNA quality and concentrations. Box plots for miRNA expression across groups is necessary.

Were all miRNAs detectable? In what % of participants? What were the Ct values?

Line 381-2: I would suggest a thorough discussion of why the results might differ across saliva microRNA studies: sex, age, race, ethnicity, saliva collection devices, RNA extraction devices, RNA quant approaches, composition of ASD/DD groups, severity of ASD group, small sample size of the current study (relative to newer studies in the field) could all be factors that led to differences across studies.

If I am interpreting the text correctly, 47% of children with ASD received a clinical evaluation. This is a limitation that should be discussed.

Reviewer #3: Sehovic et al. present work detailing the use of miRNA in saliva to distinguish between children with developmental disabilities and autism against typically developing children. Using qPCR assay the group evaluates differentially expressed miRNA and from this subset with logistic regression to find the most predictive miRNA. Further they use PLS-DA in order to work with the full set of miRNA to determine optimal sets of miRNA that are best predictive (the study draws its 14 candidate miRNA from previously published work from Hicks et al.). After iterations of modelling, the group demonstrates strong performance with sets of 5, 7, and 5 miRNA for distinguishing ASD from TD, any DD from TD, and non-ASD DD from TD, while the ability to distinguish ASD from other DD was prone to false positives.

While the idea of using salivary miRNA is not new and the authors are very clear they are drawing from sets of miRNA already implicated in earlier work, this study is important as it confirms their strength as diagnostic makers for ASD/DD. Furthermore, the use of an understudied cohort is valuable to show that these markers have widespread utility, and as the authors point out, provide accessibility in low and mid-income countries to reach necessary individuals for early intervention. Also, the paper shows the importance of looking at panels of markers rather than individual markers. Overall, the paper is well written, but would be strengthened with further presentation of the data in the paper.

While all results are displayed in the text it is difficult to gauge which are common and how great changes are by comparing sections of text. Unfortunately, the naming of miRNA does not lend itself well to easy distinction at a glance. The paper would be aided by separating DE miRNA into tables that include directional fold change differences and p-values. I realize this adds cost of redundancy, but that is offset by the value of the clarity. Tables for DE results and then for prediction results of both single logistic and PLS-DA results would be useful. I believe a heatmap showing the specific DE miRNA across all the samples and labelled with the category would be useful to see how consistent the results are across the cohort. As it stands, the buildup to the lone figure is somewhat lost, but important to understand the strength of the marker set used and the diversity of the sample cohort.

There seems to also be some clarification needed in certain areas. For instance, it was unclear why only 26 children have clinical opinion from the child psychologist and initial thoughts were that the rest were deemed non-ASD or non-DD. It is clarified that only a subset of these children were evaluated, but the message was that of agreement between CARS-II and the clinical assessment. Perhaps the table can call that section Concordance between Child Psychiatrist and CARS-II or have that added to be more clear?

It is not clear how samples were separated into training and validation. There is a line at the end of the results section that mentions 5 samples in the validation being too small for meaningful conclusions. Is that the number for all or just the non-ASD DD vs TD? The number in each set should be provided to understand the design and relevance of the training and validation sets.

Please ensure all miRNA names are miR- and not miRNA-. Also watch for some uses of mir-.

Lines 388-389 – The statement “Other notable genes were CACNA1C and DST.” is unclear. DST is already mentioned on line 385. Four target genes for the 13 miRNA with 7 target sites are given. What is notable and different about these two genes? What is different about DST mentioned here compared to line 385?

Reviewer #4: Sehovic et al. evaluated whether miRNA can be used as biomarkers in children with suspected developmental disorders within Bosnian-Herzegovinian population. They analyzed 14 selected miRNAs and identified 6 were differentially expressed between typically developing children and children with developmental disorder. Further logistic regression identified 5 as potential biomarkers for Autism spectrum disorder detection.

1. The abbreviation should be spelled out for the first time use, e.g. What’s TD in abstract?

2. Please clarify how 80 participants were sampled. Are they all the patients collected in a certain period of time?

3. 26 children were selected for clinical observations. How were these children selected? Also, how were their characteristics compared to the reaming children?

4. Please provide the reasoning or support to use z-score of 1.5 as an outlier threshold, which seems quite small.

5. Line 213 reads, “used to describe the dependency of the dependent outcome, the diagnosis in this case,..” please clarify the outcome, ASD vs DD, without considering TD? If so, it implies that the analysis of interest only focus on ASD and DD sample. Then it would be useful to present Table 2 stratified by diagnosis status rather than pooling them together. If not, then please be clear about your modeling.

6. Lines 282 and 283 talk about training set and the validation set. However, it doesn’t mention anything about different dataset in the statistical analysis section. Please provide the detail and define your training and testing datasets!

6. PLOS authors have the option to publish the peer review history of their article (what does this mean?). If published, this will include your full peer review and any attached files.

Reviewer #1: No

Reviewer #2: Yes: Steven Hicks, MD, PhD

Reviewer #3: No

Reviewer #4: No

---

## [Author Response · Author response to Decision Letter 0]

2 Feb 2020

Dear Reviewers,

Thank you so much for your time and such thorough comments and suggestions. All of your suggestions and comments have been addressed and the manuscript has been updated accordingly. Answers to your comments and how they have been addressed, in detail, please find below.

Once more thank you.

Sincerely,

Authors

REVIEWER 1

1. It is unclear how the 14 miRNAs were selected. It seems to be based on reference 49 (A comparative review of microRNA expression patterns in autism spectrum disorder. Frontiers in psychiatry. 2016); however, I do not see much overlap with the 27 miRNAs discussed in that paper.

We have selected the 14 miRNAs based on their performance in Hicks et al., 2016.

2. The validation set” was mentioned, but it is unclear what that is.

Our validation set was made up of 10% of randomly selected samples from the overall analysed cohort. 

REVIEWER 2

1. The authors are missing a key citation, which involves the largest study of saliva microRNA in children with autism (PMID: 30926572).

As suggested we have included the key citation PMID: 30926572 within the discussion section (line: 100, 460).

2. This is why additional studies (e.g. citation 54) have moved to poly-omic analysis of both microRNA and other human and microbial RNAs. For this reason, a study of <100 children is not really a validation study - especially since the authors have used different saliva collection kits, different RNA extraction techniques, and different RNA quantification protocols. The discussion does not include any mention of these factors. There is also no discussion of the study's limitations.

Poliomic analysis has been addressed and included in discussion. We agree that this is not a validation study, it is an additional test for assessment of children with developmental disorders and have made contextual changes accordingly (line: 37)

Study limitations have been addressed, (line: 425)

3. The intro would benefit from a discussion of the advantages to saliva relative to blood collection.

As suggested the advantages of salivary relative to blood collection were discussed in the introduction (line: 94)

4. This study does not show that microRNAs can be used as early as birth for ASD diagnosis. There are studies to show this type of test is not even a priority for parents of children with ASD (PMID: 30903561).

Based on the suggestion we have adjusted the utility of miRNAs as diagnostic tools as early as birth.

5. I worry that the sample size (n=25 for controls) does not support separate training/test sets. A power analysis would be useful here.

Power analysis has been performed and is included within the Supplementary Material 2.

6. Table 1 needs to compare sex/age across groups (were there differences?). Also, details about diet, oral hygiene, and oral meds would help with interpretation of the findings.

We have adjusted table 1 according to the suggestions for the control group and we have added the suggested information (sex/age) regarding DD children in Table 2. We do not have information regarding diet, oral hygiene and oral meds.

7. Were any participants excluded for inability to expectorate

There was one sample mentioned in Materials and Methods section (line: 150)

8. Please provide data on RNA quality and concentrations. Box plots for miRNA expression across groups is necessary.

Data on RNA quality and concentrations is provided to the reviewer attached as an excel file (“Samples ASD”) and the box plots for miRNA expression across groups has been added as a figure in the manuscript (Figure 1).

9. Were all miRNAs detectable? In what % of participants? What were the Ct values?

Data regarding the percentages of detected miRNAs have been added and can be found in Supplementary file 1. Raw Ct value data has been attached as an excel file (“Ct value and normalization”)

10. Line 381-2: I would suggest a thorough discussion of why the results might differ across saliva microRNA studies: sex, age, race, ethnicity, saliva collection devices, RNA extraction devices, RNA quant approaches, composition of ASD/DD groups, severity of ASD group, small sample size of the current study (relative to newer studies in the field) could all be factors that led to differences across studies.

According the suggestion we have addressed the mentioned differences in the discussion (427)

11. If I am interpreting the text correctly, 47% of children with ASD received a clinical evaluation. This is a limitation that should be discussed.

The mentioned limitation has been addressed in the discussion (line: 452)

REVIEWER 3

1. Overall, the paper is well written, but would be strengthened with further presentation of the data in the paper.

As suggested by the reviewer, additional data was added in the paper through Figure 1, Table 3 and Supplementary Tables 1 to 11 a and 2.

2. While all results are displayed in the text it is difficult to gauge which are common and how great changes are by comparing sections of text. Unfortunately, the naming of miRNA does not lend itself well to easy distinction at a glance. The paper would be aided by separating DE miRNA into tables that include directional fold change differences and p-values. I realize this adds cost of redundancy, but that is offset by the value of the clarity. Tables for DE results and then for prediction results of both single logistic and PLS-DA results would be useful. I believe a heatmap showing the specific DE miRNA across all the samples and labelled with the category would be useful to see how consistent the results are across the cohort. As it stands, the buildup to the lone figure is somewhat lost, but important to understand the strength of the marker set used and the diversity of the sample cohort.

The naming of miRNAs was addressed. Differentially expressed miRNAs with directional fold changes and p-values were included in Table 3. Regarding the suggested tables for individual miRNA LR performance and PLS-DA performance we have decided that it would be more suitable to add such tables as supplements to text in order to preserve the consistency of data presented in the text.

After constructing boxplots proposed by Reviewer 2 and heatmaps by Reviewer 3 for visualizing the expression patterns of individual miRNAs within studies groups, we have decided that boxplots visually offer more clarity and information. 

Table showing differential expression is added

3. There seems to also be some clarification needed in certain areas. For instance, it was unclear why only 26 children have clinical opinion from the child psychologist and initial thoughts were that the rest were deemed non-ASD or non-DD. It is clarified that only a subset of these children were evaluated, but the message was that of agreement between CARS-II and the clinical assessment. Perhaps the table can call that section Concordance between Child Psychiatrist and CARS-II or have that added to be more clear?

Clinical evaluation and the mentioned issues by the reviewer were thoroughly addressed in materials and methods and discussion sections of the manuscript. Concordance between child psychiatrist and CARS 2 has been added in Table 2.

4. It is not clear how samples were separated into training and validation. There is a line at the end of the results section that mentions 5 samples in the validation being too small for meaningful conclusions. Is that the number for all or just the non-ASD DD vs TD? The number in each set should be provided to understand the design and relevance of the training and validation sets.

The used program randomly separates the training and validation sets, multiple iterations were performed, and the results did not differ significantly. A smaller validation subset was selected only for non-ASD DD because their sample size was smaller than others. Based on your suggestion the whole process of validation has been more thoroughly explained within the materials and methods section of the manuscript.

5. Please ensure all miRNA names are miR- and not miRNA-. Also watch for some uses of mir-.

Based on your suggestion the naming of miRNAs has been addressed.

6. Lines 388-389 – The statement “Other notable genes were CACNA1C and DST.” is unclear. DST is already mentioned on line 385. Four target genes for the 13 miRNA with 7 target sites are given. What is notable and different about these two genes? What is different about DST mentioned here compared to line 385?

We agree with your suggestion and have removed lines 388-389 as they do not provide novel conclusions.

REVIEWER 4

1. The abbreviation should be spelled out for the first time use, e.g. What’s TD in abstract?

Based on your suggestion the abbreviation has been addressed.

2. Please clarify how 80 participants were sampled. Are they all the patients collected in a certain period of time?

The issues with sampling addressed by the reviewer have been addressed in Materials and Methods section of the manuscript (Paragraph 2)

3. 26 children were selected for clinical observations. How were these children selected? Also, how were their characteristics compared to the remaining children?

Clinical evaluation and the mentioned issues by the reviewer were thoroughly addressed in materials and methods and discussion sections of the manuscript. Concordance between child psychiatrist and CARS 2 has been added in Table 2.

4. Please provide the reasoning or support to use z-score of 1.5 as an outlier threshold, which seems quite small. ?

The reasoning behind using 1.5 Z score as the outlier threshold has been explained in materials and methods section (line: 237) 

5. Line 213 reads, “used to describe the dependency of the dependent outcome, the diagnosis in this case,..” please clarify the outcome, ASD vs DD, without considering TD? If so, it implies that the analysis of interest only focus on ASD and DD sample. Then it would be useful to present Table 2 stratified by diagnosis status rather than pooling them together. If not, then please be clear about your modeling.

According to your remark we stratified Table 2 based on diagnosis status. 

6. Lines 282 and 283 talk about training set and the validation set. However, it doesn’t mention anything about different dataset in the statistical analysis section. Please provide the detail and define your training and testing datasets!

Random separation of the training and validation sets was done and multiple iterations were performed. A smaller validation subset was selected only for non-ASD DD because their sample size was smaller than others. Based on your suggestion the whole process of validation has been more thoroughly explained within the materials and methods section of the manuscript.

---

## [Decision Letter · Decision Letter 1]

18 Feb 2020

PONE-D-19-31049R1

Identification of developmental disorders including autism spectrum disorder using salivary miRNAs in children from Bosnia and Herzegovina

PLOS ONE

Dear Dr. Hajdarpasic,

Thank you for submitting your manuscript to PLOS ONE. After careful consideration, we feel that it has merit but does not fully meet PLOS ONE’s publication criteria as it currently stands. Therefore, we invite you to submit a revised version of the manuscript that addresses the points raised during the review process.

We would appreciate receiving your revised manuscript by Apr 03 2020 11:59PM. To enhance the reproducibility of your results, we recommend that if applicable you deposit your laboratory protocols in protocols.io, where a protocol can be assigned its own identifier (DOI) such that it can be cited independently in the future. For instructions see: http://journals.plos.org/plosone/s/submission-guidelines#loc-laboratory-protocols

We look forward to receiving your revised manuscript.

Kind regards,

Lucia Billeci

Academic Editor

PLOS ONE

Reviewers' comments:

Reviewer's Responses to Questions

**Comments to the Author**

1. If the authors have adequately addressed your comments raised in a previous round of review and you feel that this manuscript is now acceptable for publication, you may indicate that here to bypass the “Comments to the Author” section, enter your conflict of interest statement in the “Confidential to Editor” section, and submit your "Accept" recommendation.

Reviewer #1: (No Response)

Reviewer #2: All comments have been addressed

Reviewer #3: (No Response)

Reviewer #4: All comments have been addressed

2. Is the manuscript technically sound, and do the data support the conclusions?

Reviewer #1: No

Reviewer #2: Yes

Reviewer #3: Yes

Reviewer #4: (No Response)

3. Has the statistical analysis been performed appropriately and rigorously? 

Reviewer #1: No

Reviewer #2: Yes

Reviewer #3: Yes

Reviewer #4: (No Response)

4. Have the authors made all data underlying the findings in their manuscript fully available?

Reviewer #1: No

Reviewer #2: Yes

Reviewer #3: Yes

Reviewer #4: (No Response)

5. Is the manuscript presented in an intelligible fashion and written in standard English?

Reviewer #1: Yes

Reviewer #2: Yes

Reviewer #3: Yes

Reviewer #4: (No Response)

6. Review Comments to the Author

Reviewer #1: 1) “We have selected the 14 miRNAs based on their performance in Hicks et al., 2016.”— It is still unclear how the 14 miRNAs were selected. The main point of Hicks et al’s paper is 27 miRNAs with overlap across ASD studies; however, I do not see much overlap with the 27 miRNAs discussed in that paper.

2) It seems the authors are not even familiar with the statistical techniques you used in the manuscript. Cross validation (CV) was used because there is no replication study; however, this key word was not even clearly written.

Reviewer #2: The authors have adequately addressed each of my comments.

The manuscript is suitable for publication.

Reviewer #3: Sehovic and colleagues have made important additions, changes and clarifications to their manuscript describing the study of miRNA to distinguish ASD and DD from typically developing children in B-H.

Additional data/results help support the evaluation of the miRNA panel and determination of optimally predictive subsets used in this study. Limitations of the study are generally made clear, so that overall context is better understood. However, it seems that there may be confusion related to the training and validation sets, or at least the terminology used, which still may need clarification.

Specifically, it appears as though the entire cohort was used as training less ~10% held back for validation. This was repeated through ‘multiple iterations’ (line 258) using the software. I would contend that this would generally be recognized as cross-validation and the paper should be clear at some point that validation is referring to cross validation, which is different from a fully withheld and independent validation set. Similar, a limitation outlining the need for a fully independent sample cohort for validation should be included.

The change of using the term spit to expectorant/expectorate is improper as it implies collection of phlegm and mucus from the lungs or throat rather than saliva (which it is meant to be reflected). This should be changed back to the previous wording (lines 107, 152).

Other corrections are required in the tables in S6, S8-S11 with respect to formatting and decimals.

Reviewer #4: (No Response)

7. PLOS authors have the option to publish the peer review history of their article (what does this mean?). If published, this will include your full peer review and any attached files.

Reviewer #1: No

Reviewer #2: No

Reviewer #3: No

Reviewer #4: No

---

## [Author Response · Author response to Decision Letter 1]

2 Apr 2020

Dear Reviewers,

Thank you so much for your time and such thorough comments and suggestions. All of your suggestions and comments have been addressed and the manuscript has been updated accordingly. Answers to your comments and how they have been addressed, in detail, please find below.

Once more thank you.

Sincerely,

Authors

REWIER 1

1) “We have selected the 14 miRNAs based on their performance in Hicks et al., 2016.”— It is still unclear how the 14 miRNAs were selected. The main point of Hicks et al’s paper is 27 miRNAs with overlap across ASD studies; however, I do not see much overlap with the 27 miRNAs discussed in that paper.

There are 2 articles published in 2016 by Hicks and his colleagues. The article titled “Salivary miRNA profiles identify children with autism spectrum disorder, correlate with adaptive behavior, and implicate ASD candidate genes involved in neurodevelopment” has found 14 differentially expressed miRNAs between TD and ASD children. Hence, based on their predictive performance they were chosen for analysis within the B&H population.

2) It seems the authors are not even familiar with the statistical techniques you used in the manuscript. Cross validation (CV) was used because there is no replication study; however, this key word was not even clearly written.

The issue with clarifying the terminology regarding validation was addressed. The term cross validation is now used instead of using the term validation on its own (Line: 257-258; 392-394). 

REWIER 3

1) It seems that there may be confusion related to the training and validation sets, or at least the terminology used, which still may need clarification. Specifically, it appears as though the entire cohort was used as training less ~10% held back for validation. This was repeated through ‘multiple iterations’ (line 258) using the software. I would contend that this would generally be recognized as cross-validation and the paper should be clear at some point that validation is referring to cross validation, which is different from a fully withheld and independent validation set. Similar, a limitation outlining the need for a fully independent sample cohort for validation should be included.

As suggested by the reviewer, we have addressed the issue with using the term validation without stating it is cross validation. In materials and methods, we have changed validation to cross validation and have made clear that any type of validation which is mentioned later in the manuscript should be considered as a cross validation (Line: 257-258).

As suggested by the reviewer, we have addressed the limitation of the study regarding the need for an independent sample cohort for validation (Line: 392-394)

2) The change of using the term spit to expectorant/expectorate is improper as it implies collection of phlegm and mucus from the lungs or throat rather than saliva (which it is meant to be reflected). This should be changed back to the previous wording (lines 107, 152).

As suggested by the reviewer, the wording in lines 107 and 152 was returned back to the previous wording and the word expectorate was replaced by saliva.

3) Other corrections are required in the tables in S6, S8-S11 with respect to formatting and decimals.

Formatting corrections were made in the tables mentioned by the reviewer.

Thank you.

---

## [Decision Letter · Decision Letter 2]

14 Apr 2020

Identification of developmental disorders including autism spectrum disorder using salivary miRNAs in children from Bosnia and Herzegovina

PONE-D-19-31049R2

Dear Dr. Hajdarpasic,

We are pleased to inform you that your manuscript has been judged scientifically suitable for publication and will be formally accepted for publication once it complies with all outstanding technical requirements.

With kind regards,

Lucia Billeci

Academic Editor

PLOS ONE

Additional Editor Comments (optional):

Reviewers' comments:

Reviewer's Responses to Questions

**Comments to the Author**

1. If the authors have adequately addressed your comments raised in a previous round of review and you feel that this manuscript is now acceptable for publication, you may indicate that here to bypass the “Comments to the Author” section, enter your conflict of interest statement in the “Confidential to Editor” section, and submit your "Accept" recommendation.

Reviewer #1: All comments have been addressed

Reviewer #3: All comments have been addressed

2. Is the manuscript technically sound, and do the data support the conclusions?

Reviewer #1: Yes

Reviewer #3: Yes

3. Has the statistical analysis been performed appropriately and rigorously? 

Reviewer #1: Yes

Reviewer #3: Yes

4. Have the authors made all data underlying the findings in their manuscript fully available?

Reviewer #1: Yes

Reviewer #3: Yes

5. Is the manuscript presented in an intelligible fashion and written in standard English?

Reviewer #1: Yes

Reviewer #3: Yes

6. Review Comments to the Author

Reviewer #1: There is no more comments.

There is no more comments.

There is no more comments.

There is no more comments.

Reviewer #3: The concerns have been addressed. Clarity on cross-validation has improved and readers of the text should be able to distinguish that the validation herein refers to cross-validation. For clarity, I would suggest the term validation in the abstract (line 37) is written cross-validation.

7. PLOS authors have the option to publish the peer review history of their article (what does this mean?). If published, this will include your full peer review and any attached files.

Reviewer #1: No

Reviewer #3: No

---

## [Editor Report · Acceptance letter]

17 Apr 2020

PONE-D-19-31049R2 

Identification of developmental disorders including autism spectrum disorder using salivary miRNAs in children from Bosnia and Herzegovina 

Dear Dr. Hajdarpasic:

I am pleased to inform you that your manuscript has been deemed suitable for publication in PLOS ONE. Congratulations! Your manuscript is now with our production department. 

With kind regards,

on behalf of

Dr. Lucia Billeci 

Academic Editor

PLOS ONE